# Pollination in *Epidendrum densiflorum* Hook. (Orchidaceae: Laeliinae): Fraudulent Trap-Flowers, Self-Incompatibility, and a Possible New Type of Mimicry

**DOI:** 10.3390/plants12030679

**Published:** 2023-02-03

**Authors:** Rodrigo Santtanna Silveira, Rodrigo Bustos Singer, Viviane Gianluppi Ferro

**Affiliations:** 1Programa de Pós-Graduação em Botânica, Universidade Federal do Rio Grande do Sul, Porto Alegre 91509-900, RS, Brazil; 2Departamento de Zoologia, Laboratório de Ecologia de Insetos, Instituto de Biociências, Universidade Federal do Rio Grande do Sul, Porto Alegre 91501-970, RS, Brazil

**Keywords:** Arctiidae, *Epidendrum*, Ithominae, Laeliinae, Orchidaceae, pollination

## Abstract

The pollination and the breeding system of *Epidendrum densiflorum* (Orchidaceae: Laeliinae) were studied through fieldwork and controlled pollinations in cultivated plants. Pollination is exclusively promoted by males of diurnal Lepidoptera: five species of Arctiinae and four of Ithomiinae were recorded as pollinators. These male insects are known to obtain alkaloids (through the nectar) in flowers of Asteraceae and Boraginaceae. However, the flowers of *E. densiflorum* are nectarless, despite presenting a cuniculus (a likely nectariferous cavity). Pollinators insert their proboscides into the flowers and remove or deposit the pollinaria while searching for nectar. The floral tube is very narrow, and insects struggle for up to 75 min to get rid of the flowers. Plants are pollinator-dependent and nearly fully self-incompatible. Pollinarium removal, pollination, and fruiting success (2.85%) were very low; facts that are consistent with the patterns globally observed in deceptive (rewardless) orchids. Nilsson’s male efficiency factor (0.245) was also low, indicating pollen loss in the system. Based on our field observations, we suggest that the fragrance of *E. densiflorum* likely mimics these plants that are normally used as a source of alkaloids by male Lepidoptera, a hypothesis that we intend to test in the future.

## 1. Introduction

*Epidendrum* L. (Orchidaceae: Laeliinae) ranks among the biggest Neotropical orchid genera, embracing about 1500 described species and occurring from the Southern United States to Northern Argentina [1]. Phylogenetic studies suggest that *Epidendrum* is monophyletic [1]. The genus can be easily diagnosed by its characteristic flowers, with the labellum (median petal) fused to the lateral sides of the column, forming a narrow tube [1,2]. As a result, the flower cavity is tubulose and long. In addition, as in most other orchid genera of subtribe Laeliinae, *Epidendrum* flowers present a well-developed cuniculus (a putative nectariferous cavity) below the column and parallel to the ovary [1,2]. However, in most studied *Epidendrum* species, this cuniculus is not secretory [1]. Darwin [3] was the first researcher to speculate that *Epidendrum* flowers are morphologically suitable for pollination by Lepidoptera. Darwin [3] manipulated flowers and correctly speculated that the pollinarium of these flowers should glue onto the surface of the proboscides of their pollinators. Hágsater and Soto Arenas [1] (pp. 247–250) and Van der pijl and Dodson [4] (p. 185) list and review several brief observations of floral visitors in several species. However, these reports are essentially anecdotal and mostly consist of observations of insects at flowers with no evidence that these animals (bees, moths, flies, and butterflies) remove and deposit pollinaria, the minimum requirement to consider them as pollinators [5]. Despite the size of the genus, few species have been properly studied concerning the pollination mechanism and breeding system [6,7,8,9,10], and most of these studies indicate Lepidoptera as the main pollinators, in agreement with the funnel-like overall floral structure. *Epidendrum fulgens* Brongn [6] and *Epidendrum secundum* Jacq [8] are pollinated by butterflies. Pollination by butterflies and diurnal moths (Arctiinae) was documented for *Epidendrum paniculatum* Ruiz & Pav. [7], and *Epidendrum avicula* Lindl. is reported as pollinated by both moths and Tipulidae flies [10]. Remarkably, the flowers of *Epidendrum trydactylum* Lindl. are pollinated only by flies [9]. Most of the aforementioned studies [6,7,8,9] pointed out that the species present nectarless flowers, that is, all these *Epidendrum* species may be deceptive regarding their pollination strategies. However, the flowers of *E. avicula* are referred to as nectar-secreting [10]. As a whole, all studied species are pollinator-dependent and natural fruiting success is consistently low [6,7,8,9,10]. Regarding their breeding systems, most detailed studies published so far indicate self-compatibility (plants can set fruit following self-pollination) [6,7,8,9,10] and, more rarely [7,10], self-incompatibility (plants are unable to set fruit following self-pollination and need pollen from another conspecific individual to set fruit). *Epidendrum densiflorum* is widespread in Brazil [11] and also occurs in neighboring countries such as Argentina, Colombia, and Venezuela. In Southern and Southeastern Brazil, this species is found within the Atlantic Rain Forest (Mata Atlântica) and Cerrado domains. This species has not been studied yet regarding its pollination and breeding system. Since 1995, one of us (RBS) has regularly observed Lepidoptera, visiting this species in the wild as well as under cultivation. However, fruits are rarely observed under natural conditions (RB. Singer, pers. obs). In this work, we will answer the following questions: (1) is *E. densiflorum* pollinator-dependent? (2) if it is pollinator-dependent, which animals are the pollinators and what is their behavior at flowers? (3) what is this species’ breeding system? and (4) what is this species’ fruiting success under natural conditions? According to the literature [6,7,8,9,10], as well as based on preliminary observations, we establish the following hypotheses for these questions: (1) *E. densiflorum* may be pollinator-dependent, (2) pollinators may be diurnal Lepidoptera, (3) owing to the rareness of fruits under natural conditions, we think that *E. densiflorum* may be self-incompatible, and (4) fruit rareness (as stated in 3) may be caused by either the rareness of pollinators or by self-incompatibility.

## 2. Results

### 2.1. Flower Features

Only flower features pertinent to the pollination process will be presented. Readers interested in more details of plant and flower morphology are referred to [12]. At the study locality, plants have 2–6 inflorescences, with 8–19 flowers each. The flowers are non-resupinated [2], but the lack of resupination is “corrected” by the hanging habit of the inflorescence (Figure 1). Hence, the labellum is presented as a landing platform (Figure 1). Each inflorescence has up to 19 greenish-white flowers with a diameter of ca. 25 mm each (Figure 1). The column is slightly arched and crowned by a single, terminal anther that holds a pollinarium made up of four laterally compressed, yellow, indivisible pollinia connected to a terminal, pad-like, detachable viscidium [2]. Untouched flowers keep their fresh appearance for up to seven days (*n* = 20), quickly wilting after this period. None of the flowers (*n* = 10) tested with the microsyringe presented free nectar. Another ten fresh flowers were dissected under a stereomicroscope and showed no signal of free nectar, too. Thus, we consider that *E. densiflorum* is deceitful, presenting nectarless flowers.

### 2.2. Breeding System and Fruiting Success under Natural Conditions

The results of breeding system experiments are summarized in Table 1. No fruits were obtained through intact or emasculated flowers, clearly indicating that *E. densiflorum* is pollinator-dependent and, thus, unable to set fruit in absence of pollinators. As the data analyzed between treatments did not show the normal distribution in fruit production, the Kruskal–Wallis test was used (x^2^(1) = 87.51, df = 3, *p*-value = *p* < 0.001), which showed that there are significant differences between treatments and fruit generation: cross-pollination (48 fruits, mean of 0.96 ± 0.19 per plant), self-pollination (1 fruit, mean of 0.02 ± 0.14 per plant), and intact flowers and emasculation, showing no fruit formation.

Under natural conditions (Table 2), only 28 fruits (over 969 flowers) were formed during the observation period. This represents ca. 2.88% of natural fructification. As a whole, during the observation period, 158 flowers acted as pollen donors and 41 acted as pollen receivers. Nilsson’s male efficiency factor scored 0.259 (Table 2), indicating pollen loss in the system. This value indicates that 0.259 flowers were pollinated by pollinarium removal.

### 2.3. Pollinators, Pollinator Activity, and the Pollination Process

Pollinator activity was recorded between 07:45 h and 17:30 h (Figure 2), with most visits occurring between 09:00 h and 16:00 h (Figure 2).

A total of 64 pollinator visits were recorded (Table 3). All recorded pollinators were males of diurnal Lepidoptera of subfamilies Arctiinae (Erebidae) and Ithomiinae (Nymphalidae) (Table 3 and Figure 3). Arctiinae moths were responsible for 81.25% of the observed pollination events (52 visits and 84.7% of the observed pollinarium removals). We recorded five species of tiger moths pollinating *E. densiflorum*: *Phoenicoprocta teda* Walker (26.56% of the visits and 31% of pollinarium removals), *Antichloris eriphia* Fabricius (20.31% of visits and 20% of pollinarium removals), *Philoros rubriceps* Walker (17.18% of visits and 17% of pollinarium removals), *Calodesma collaris* Drury (14% of visits and 15% of pollinarium removals), and *Cyanopepla jucunda* Walker com (3.12% of visits and 2% of pollinarium removals). Among the Ithominae, we recorded four species: *Methona themisto* Hübner (7.81% of visits and 7% of pollinarium removals), *Episcada hymenaea* Prittwitz (6.25% of visits and 4% of pollinarium removals), *Hypothyris euclea* Doubleday (3.12% of visits and 2% of pollinarium removals), and *Placidina euryanassa* C. & R. Felder (1.5% of visits and 2% of pollinarium removals).

The pollination mechanism is the same, irrespective of the insect species involved (Figure 4 and Appendix A): moths and butterflies insert the proboscis inside the floral tube looking for nectar. After a variable period (see below, Table 3) of struggling, these Lepidoptera withdraw the proboscis carrying the pollinarium adhered on its surface (Figure 4). The pad-like viscidium of the pollinarium is responsible for its adherence to the proboscis. The pollinarium is removed with the anther cap, which falls after a few seconds (Figure 4). Pollination takes place when a pollinarium-carrying insect visits another flower and the pollinia are arrested at the flower’s hollow stigmatic surface. Insects removing pollinaria from fresh flowers spend (depending on the species) 4–73 min struggling to get rid of the flowers. The mean time of permanence in the flowers is shown in Table 3. The results showed significant differences between the mean time of the visitation of the species of the Arctinae and Ithomiinae subfamilies, (Kruskal–Wallis X2 = 31.8, df = 8, *p*-value < 0.001), however, they did not present significant differences when comparing the values of the mean time of visitation of individuals of the same subfamily. Itomiinae visits are significantly faster, with an overall average visitation of (7.5 min), compared to 55.8 min of Arctiinae. During our observations, we only recorded Arctiinae moths of *Antichloris eriphia* and *Phoenicoprocta teda* laden with pollinaria, depositing them at the stigmatic cavities of flowers.

## 3. Discussion

### 3.1. Flower Features

Overall, recorded flower features are in agreement with those already mentioned in the literature for other *Epidendrum* species [6,7,8] or Laeliinae orchids as well [2]. As in most Laeliinae orchids, the flowers of *Epidendrum densiflorum* present a well-developed cuniculus [2], a structure that has been interpreted as a nectariferous cavity. However, researchers have often reported that, despite the presence of the cuniculus, no free nectar is found in Laeliinae flowers [1,13,14] and, in agreement with the absence of nectar, pollination events are rare [6,13,14], as seen below. Based on anatomical features alone, Cardoso-Gustavson et al. [15] have challenged this idea and have proposed that many *Epidendrum* species (including *E. densiflorum*) are nectar-secreting. Conversely, during our observations, we did not find any free nectar in the flowers of *E. densiflorum* and, in agreement with this, we recorded a low visitation rate, low rates of pollinarium removal and deposition, and a low fruiting success (see below), which are all consistent features among deceptive orchids [16,17]. When compared to rewardless orchids, rewarding ones almost score double fruiting success [16]. Pansarin [7] also did not find free nectar at the cuniculus of the closely-related *E. paniculatum*. Some researchers have speculated [4] that the cells within the cuniculus may be thin-walled and may easily break, exposing a liquid content when contacted by the mouthparts of the pollinators. However, this has not been proven yet, and the values of reproductive success (see below) in *Epidendrum* species studied so far are consistent with those of nectarless/deceptive orchid species [16,17]. We would like to stress that, whereas Cardoso-Gustavson et al. [15] made their proposal of nectar secretion based on detailed anatomical studies, they did not give any details of nectar features, such as volume and concentration, that are frequently mentioned in the literature [18,19]. Orchids pollinated by Lepidoptera normally present a nectar column inside their nectar spurs or nectariferous cavities, and the properties of nectar (volume, concentration) are quantifiable [18].

### 3.2. Breeding System and Fruiting Success under Natural Conditions

Most Laeliinae orchids are self-compatible [14,20,21,22,23,24], being able to set fruit following self-pollination. However, *E. secundum* [8], *E. fulgens* [6], and *E. tridactylum* [9] are self-compatible, but also pollinator-dependent, being unable to set fruit in absence of pollinators. In our study, flowers of *E. densiflorum* also behaved as pollinator-dependent and were almost completely self-incompatible, aborting most (98%) of the hand self-pollinations. Conversely, these plants had a high fruit development through cross-pollination (95%). Very similar results were found by Pansarin [7] while studying the breeding system of the closely-related *E. paniculatum*. Self-incompatibility was also found in the nectar-secreting *E. avicula* [10]. The latter species is very morphologically different from *E. paniculatum* and *E. densiflorum*, a fact that suggests that self-incompatibility may have evolved more than once in the genus. It is important to point out, however, that, even in self-compatible Laeliinae orchids, the number of viable seeds obtained from cross-pollination can be significantly higher when compared with the results of self-pollination [14,22,23].

Under natural conditions, pollinarium removal, deposition, and fructification were low. These results concur with those obtained by Pansarin [7] for *E. paniculatum*. Nilsson’s male efficiency factor was also low (0.25). Overall, this value indicates that one in four dislodged pollinaria reached a stigmatic cavity. The low fruiting success (less than 3%) is, in our opinion, explained by the following factors: (1) low pollinarium removal and deposition and (2) self-incompatibility. By comparing male (16.3% of available flowers) and female functions (4.2%), it is possible to notice that roughly one-fourth of the pollinated flowers turned into fruits. This, in our opinion, can be partially explained by self-incompatibility: possibly, some pollinarium-laden butterflies returned to the plants from where they dislodged the pollinaria and promoted some insect-mediated self-pollinations. It is well known that males of Ithomiinae butterflies are territorial [25], thus, it seems possible that such behavior (in addition to self-incompatibility) could prompt insect-mediated self-pollinations and, ultimately, abortions. Analog scenarios have already been proposed for other Epidendroids orchids with self-incompatibility and low natural fruit sets [26,27].

Since we did not find free nectar at the flower’s cuniculus, we suggested that *E. densiflorum* is a deceptive orchid. The observed low fruiting success is in full agreement with this proposal. Fruiting success in Orchidaceae has been reviewed by Tremblay et al. [16] and Neiland and Wilcock [17]. As a whole, fruiting success in Orchidaceae roughly surpasses 17%, but is especially low in deceptive (rewardless) orchids, ranging from less than 1 to 7% [16,17]. Moreover, the obtained value for Nilsson’s male efficiency factor indicates that only a fraction of the available flowers participated in reproduction and that there was pollen loss in the system.

### 3.3. Pollinators and Pollinator Behavior

This contribution confirms the importance of Lepidoptera as *Epidendrum* pollinators [4,7,8,10]. The characteristic flower structure, with the lateral sides of the column fused to the labellum, makes *Epidendrum* flowers particularly suitable for Lepidoptera, since the narrow floral tube only allows the entrance of their proboscises [2,4]. During this study, we found males of Arctiinae and Ithomiinae as pollinators, and this is in full agreement with preceding reports on closely-related species, such as *Epidendrum floribundum* and *E. paniculatum* [7,28,29], which also found pollinators of the same taxonomic groups. Despite being taxonomically distant, males of Ithomiinae and Arctiinae share an important ecological feature: males of both groups actively collect pyrrolizidine alkaloids that are used during mating [30]. Alkaloids are acquired through floral nectar and, according to Pliske [30], the main sources of these alkaloids involve species of the genera *Heliotropium*, *Tournefortia*, and *Myosotis* (Boraginaceae), as well as *Eupatorium* (Asteraceae). Early observations of Arctiinae and Ithomiinae in species related to *E. densiflorum*, such as *E. floribundum* and *E. paniculatum* [28], led researchers to propose that the flowers of these orchids should also be a source of pyrrolizidine alkaloids. However, alkaloids were not found in the flowers of these species [7]. According to our observations, flowers of *E. densiflorum* are devoid of free nectar, as the related *E. paniculatum* [7]. Since alkaloids are acquired through nectar, and the latter is absent in the flowers of *E. densiflorum*, there is no way for the male Lepidoptera to obtain such resources in the flowers of the orchid under study. Flowers used for breeding system experiments were enclosed with white tule (see Methods). During the whole flowering season of 2021 and 2022, the enclosed flowers were regularly visited by Arctiinae and Ithomiinae males (Figure 5). This behavior is similar to that described by De Vries and Stiles [28], which noticed males of Arctiinae and Ithomiinae attempting to reach enclosed inflorescences of *E. paniculatum*. This highlights the importance of fragrance features (chemical compounds) in this pollination strategy. Since the used covering is white, flower color and shape features are hidden. Thus, we assume for now that the observed attraction is mostly or mainly mediated by fragrance volatiles, a common feature in deceptive orchids [31]. Finally, we would like to propose a new hypothesis for the pollination strategy of *E. densiflorum*: instead of offering alkaloids, we propose that the flowers of *E. densiflorum* (and related species) mimic the fragrances of plants that are the actual alkaloid sources for these Lepidoptera. If this hypothesis is correct, it may constitute a specialized, novel subtype of pharmaco-food mimicry in Orchidaceae, since the flowers may explore a narrow, specialized, ecological niche.

Whereas pollination by Lepidopterans was already documented in a few *Epidendrum* species [7,8,10], the fact that these insects are temporarily trapped by the flowers was largely overlooked. According to our observations, insects can remain on the flower for more than 75 min, depending on the species. Our video record indicates that, after struggling to leave the flower, pollinarium-laden insects fly away without visiting another flower of the inflorescence. Such behavior may increase the chances of cross-pollination. However, it is important to point out that some small Arctiinae moths died at the flowers, resembling what happens with the “moth catcher” vine, *Araujia sericifera* (Apocynaceae) [32]. This last observation suggested that the flowers of *E. densiflorum* can be “trap-flowers”. The presence of trap-flowers in Orchidaceae has already been documented in the subfamily Cypripedioideae (reviewed by [33]) and in the pseudocopulatory genera *Pterostylis* (Orchidoideae) [34] and *Trigonidium* (Epidendroideae: Maxillariinae) [35]. In none of the above cases, however, the trapping involves nectar-seeking insects, such as in *E. densiflorum*. Flowers of Cypripedioideae are food or oviposition site mimics (reviewed by [33]). The flowers of *Pterostylis* and *Trigonidium* attract insect males (of Sciariidae flies and Meliponina bees, respectively) that attempt copulation with the median petal (labellum) and are then temporarily trapped in the flower cavity during the process [34,35].

## 4. Materials and Methods

### 4.1. Study Place

Field observations were performed at the Parque Municipal das Oito Cachoeiras (Municipality of São Francisco de Paula, Rio Grande do Sul State, Brazil (29°28″ S e 50°31″ W). The Park comprises about 136.74 hectares and is inserted within the Atlantic Rain Forest Domain. The altitude varies from 620 to 890 m and rainfall frequently exceeds 2000 mm/year [36].

### 4.2. Study Species and Flower Features

Throughout this contribution, overall orchid morphological terms follow Dressler [2]. Taxonomic delimitation of Brazilian *Epidendrum* species follows Pessoa [11]. *Epidendrum densiflorum* locally occurs both as an epiphytic or rupicolous species, always near river courses and streams. The cane-like stems may reach up to 1 m high [12]. The inflorescences are terminal and may reach up to 40 cm in length. The species has a flowering peak in January to April. A plant voucher is deposited at the ICN Herbarium (R. B. Singer s.n. 19/05/2019). Plant and overall flower features were documented through photos. To locate nectar, ten buds were isolated with tulle and checked for nectar 24 h after opening. A microsyringe CG model 701 RN (5 µL volume) was inserted in the floral cuniculus to pump the nectar (if present). Since no free nectar was found (see Results, Section 2), no further analyses (volume, concentration) were possible. Complementary, ten additional fresh, intact flowers were obtained from two specimens and dissected under a stereomicroscope to locate nectar.

### 4.3. Fieldwork Observations

Field observations were performed from 26 February 2022 to 16 March 2022, totaling 144 observation hours. As a whole, 16 plants, bearing 67 inflorescences and 969 flowers, were monitored. Preliminary observations in cultivated plants indicated that the flowers emit fragrances during the day. Therefore, observations were performed from 07:30 h to 18:30. Pollinators and their behaviors at flowers were documented through photos and videos. The video record was useful to record pollinator behavior as well as to confirm which insects were pollinators (see below) and to quantify the time spent by the insects at the flowers. For the purposes of this contribution, and following Castro et al. [27], only animals that were observed removing and/or depositing pollinaria were considered pollinators. During the whole observation period, we recorded the percentages of pollinarium removals (male success) and depositions (female success) over the total of produced flowers. The average pollinator visitation time was compared using the Krukal–Wallis test, a non-parametric alternative, since the data did not meet the assumptions of normality and homoscedasticity of variances required by analysis of variance. Dunn’s test was performed to differentiate a posteriori means with Holm’s correction [37]. The ggstatsplot and ggplot2 packages were used for the analyses and performed in the R 4.0.5 software [38]. In addition, fruiting success was recorded in 10 plants bearing 437 flowers, by counting the number of fruits formed in their inflorescences 15 days after the end of the pollination observations. Additionally, we calculated Nilsson’s male efficiency factor as the ratio between the percentages of pollinated flowers divided by the percentage of flowers that acted as pollen donors [39]. Pollinators were identified, and vouchers were deposited at the Museu de Ciências Naturais do Jardim Botânico de Porto Alegre (MCN, SEMA, RS).

### 4.4. Breeding System

The breeding system of *E. densiflorum* was studied by employing ten individuals cultivated at the Orchidarium of the Porto Alegre Botanical Garden, following the procedures detailed by Castro et al. [27]. Inflorescences were covered with tule to avoid pollinators and/or other insects. Four treatments were applied (Table 1): (1) intact flowers (test for autonomous self-pollination), (2) emasculation (test for apomixis), (3) hand self-pollination (test for self-compatibility), and (4) cross-pollination. The four treatments were applied to all the individuals under study. A total of 5 replications of each treatment were performed in each specimen, totaling 50 replications per treatment (Table 1). The numbers of fruits were compared between treatments using the Krukal–Wallis test, a non-parametric alternative, since the data did not meet the assumptions of normality and homoscedasticity of variances required by analysis of variance (Anova). Dunn’s test was performed to differentiate a posteriori means with Holm’s correction [37]. The ggstatsplot and ggplot2 packages were used for the analyses and performed in the R 4.0.5 software [38].

## 5. Conclusions

At the end of the Introduction, we proposed the following hypotheses: (1) that *E. densiflorum* may be pollinator-dependent, (2) that pollinators may be diurnal Lepidoptera, (3) that, owing to the rareness of fruits under natural conditions, *E. densiflorum* may be self-incompatible, and (4) that fruit rareness may be caused by either absence of pollinators or by self-incompatibility. According to the data gathered in this contribution, hypotheses (1–3) were fully corroborated and hypothesis (4) receives at least partial support. Pollinators were present and the plant was, indeed, self-incompatible. However, further observations are needed to fully address the extent of insect-mediated self-pollinations (and, consequently, abortions mediated by passive or territorial insects) are necessary. Unlike other Epidendroid orchids recently studied (e.g., [27]), the pollinators of *E. densiflorum* tend to visit a single flower at the inflorescences. The exhaustion after removing the pollinarium from the trap-flowers of *E. densiflorum* makes the insects quickly leave the inflorescences. However, it cannot be discarded that insects could return later. This is especially true for territorial Lepidoptera, such as the Ithomiinae. As already commented by other authors [27], pollinator-dependent, self-incompatible orchids (such as *E. densiflorum*) may be particularly fragile in the context of habitat fragmentation, since pollen flux between conspecific individuals is mandatory. Thus, studies such as those presented herein are important if these orchids and their pollinators are to be conserved and correctly managed.

## Figures and Tables

**Figure 1 plants-12-00679-f001:**
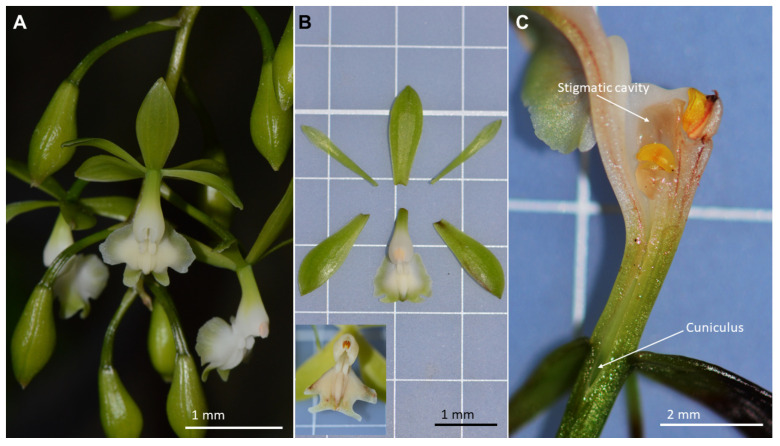
Overall morphological features of *Epidendrum densiflorum* Hook. (**A**) Inflorescence. (**B**) Dissected perianth and detail of column. (**C**) Flower in longitudinal section, showing the concave stigmatic cavity and the empty cuniculus.

**Figure 2 plants-12-00679-f002:**
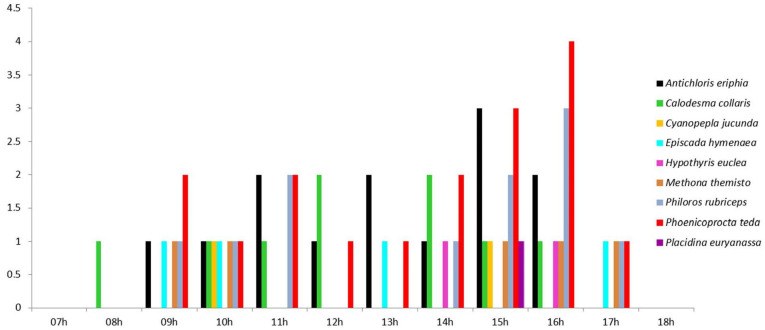
The number of lepidopteran pollinators (vertical axis) that visited *Epidendrum densiflorum* flowers throughout the day (horizontal axis).

**Figure 3 plants-12-00679-f003:**
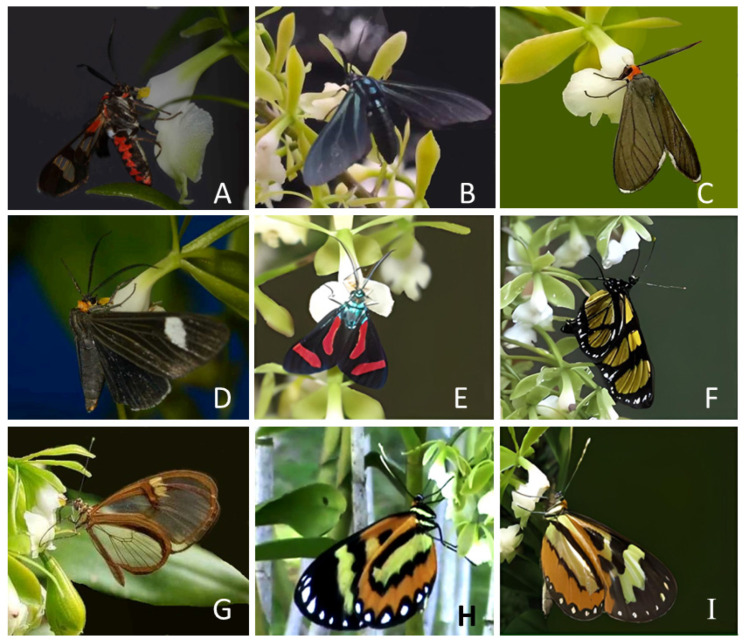
Diversity of the observed *Epidendrum densiflorum* pollinators. (**A**) *Phoenicoprocta teda*; (**B**) *Antichloris eriphia*; (**C**) *Philoros rubriceps*; (**D**) *Calodesma collaris*; (**E**) *Cyanopepla jucunda*; (**F**) *Methona themisto*; (**G**) *Episcada hymenaea*; (**H**) *Placidina euryanassa*; (**I**) *Hypothyris euclea*.

**Figure 4 plants-12-00679-f004:**
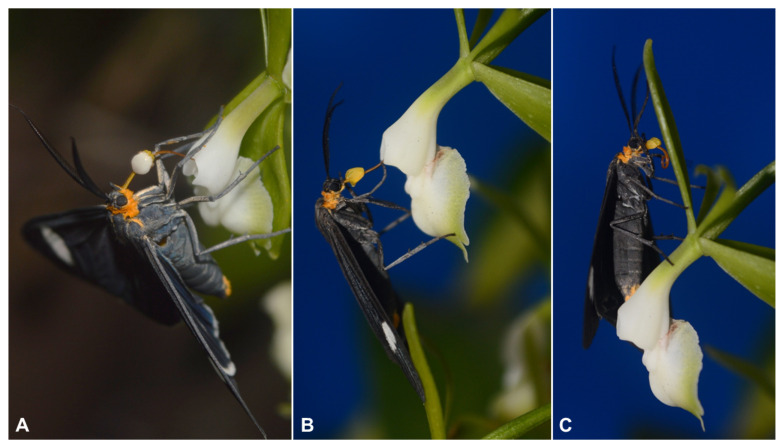
Process of pollinarium withdrawal by Arctiinae moth (*Calodesma collaris*). (**A**) The just-removed pollinarium still holds the anther cap. (**B**) After a few seconds, the anther cap falls and (**C**) the moth leaves the inflorescence.

**Figure 5 plants-12-00679-f005:**
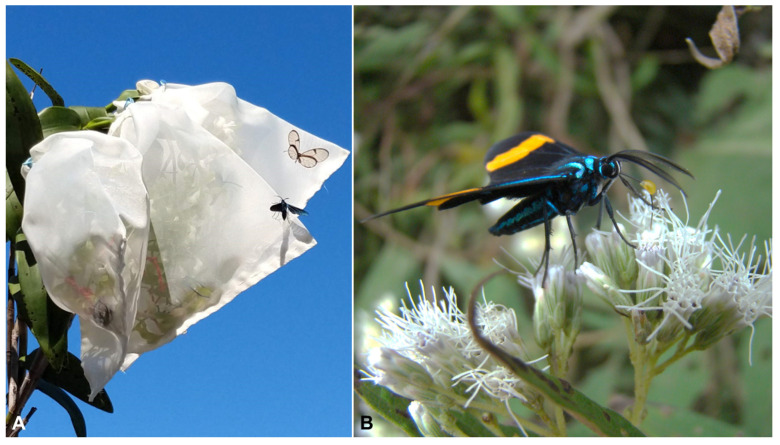
(**A**) Arctiinae moth and Ithomiinae butterflies attracted to an enclosed specimen of *E. densiflorum*. (**B**) Pollinarium-laden Arctiinae moth visiting inflorescences of *Mikania* sp. (Asteraceae), recorded in April 2009 at the Parque Nacional Aparados da Serra (Cambará do Sul Municipality, RS, Brazil).

**Table 1 plants-12-00679-t001:** Tests of the reproductive system of *Epidendrum densiflorum* Hook. Mean values followed by the distinct letter ⁽ᵃ^,b^⁾ indicate a significant difference (*p* < 0.05) according to the pairwise Dunn test. DP = standard deviation and SE = standard error.

Treatments N = 50 Flowers/Treatment	Fruits			
	Total	Average	SE	SD
Cross-pollination	48	0.96 ^a^	0.028	0.198
Self-pollination	1	0.02 ᵇ	0.020	0.141
Emasculation	0	0 ᵇ	0.0	0.00
Intact flowers	0	0 ᵇ	0.0	0.00

**Table 2 plants-12-00679-t002:** Reproductive success, fruit set, and Nilsson efficiency factor under natural conditions in *Epidendrum densiflorum* Hook.

Pollination Success (Open Pollination) N = 969 Flowers	Number/Percentage
Pollinaria removed	158 (16.30%)
Pollinaria deposited	41 (4.23%)
Nilsson’s male efficiency factor	0.259%
Fructification	28 (2.88%)

**Table 3 plants-12-00679-t003:** Number of visits, sex of the specimen, number of pollinaria removed, and average time (measured in minutes) of permanence in the flowers, followed by SD = standard deviation and SE = standard error of each species of lepidopteran pollinator of *Epidendrum densiflorum* Hook. The mean values followed by the distinct letter ⁽ᵃ⁻ᵇ⁾ indicate a significant difference (*p* < 0.05) according to the paired Dunn test.

Family/Subfamily Species	Visits	Gender	Pollinaria Removed	Average Time in Flower (min)	SD	SE
**Erebidae: Arctiinae**						
*Antichloris eriphia* Fabricius, 1776	13	male	9	47 ᵃ	13.9	4.55
*Calodesma collaris* Drury, 1782	9	male	7	62 ᵃ	8.3	10.1
*Cyanopepla jucunda* Walker, 1854	2	male	1	38 ᵃ	6.7	4.00
*Philoros rubriceps* Walker, 1854	11	male	8	59 ᵃ	10.4	8.63
*Phoenicoprocta teda* Walker, 1845	17	male	14	73 ᵃ	16.5	10.4
**Nymphalidae: Ithomiinae**						
*Episcada hymenaea* Prittwitz, 1865	4	male	2	15 ᵇ	7.1	3.15
*Hypothyris euclea* Doubleday, 1847	2	male	1	4 ᵇ	2.4	1.00
*Methona themisto* Hübner, 1818	5	male	3	6 ᵇ	5.2	0.70
*Placidina euryanassa* C. & R. Felder, 1860	1	male	1	5 ᵇ	0	0

## Data Availability

Data available on request, from the correspondent author.

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
