# Peer review of "Pollination in Epidendrum densiflorum Hook. (Orchidaceae: Laeliinae): Fraudulent Trap-Flowers, Self-Incompatibility, and a Possible New Type of Mimicry"

_plants, 2023, doi:10.3390/plants12030679_

Round 1
Reviewer 1 Report
I was pleased to read the manuscript of the work entitled Pollination in Epidendrum densiflorum Hook. (Orchidaceae: Laeliinae): fraudulent trap-flowers, self-incompatibility, and a possible new type of mimicry involving males of Lepidoptera. The text is written clearly, both in relation to the presented assumptions of the work and hypotheses. The results were well described and provide a new information on pollination of Epidendrum densiflorum. In the Discussion part, the authors referred to all the issues studied. The mimicry hypothesis seems very interesting and highly probable, considering that only males visited the flowers. However, I wonder if it was the white insulators could be attractants for pollinators. I would suggest using in the next experiment insulators of cryptic colour on the inflorescences, for comparison with white ones. And as a control, to use white and dark ‘empty’ insulators.
Minor comments are marked on attached pdf

Author Response
Answers to reviewer 1:
I warmly acknowledge your suggestions/corrections that have certainly improved the manuscript. I have followed them as close as possible. Otherwise than indicated, I modified the text according to the suggestions/corrections you sent via PDF. Modifications (including those suggested by another reviewer) are highlighted in yellow, to make the checking easier. On only one occasion I couldn't follow your suggestions: in the PDF (line 323) you ask for the voucher numbers of insect pollinators. This is not possible, because at the MCN/SEMA/RS the insects are grouped by taxonomic categories, but do not receive a number (at least, not at the present). In addition, and following your suggestion on the color of the tulle as a potential attractant, I slightly modified the text (see Line 258). We will keep this suggestion in mind for future bioassays, thank you very much. Other than this, the manuscript was modified according to all your suggestions. Thank you sincerely for your time and attention,
Prof. Rodrigo B. Singer (correspondent author)

Reviewer 2 Report
I read with great interest the article titled: 'Pollination in Epidendrum densiflorum Hook. (Orchidaceae: Laeliinae): fraudulent trap-flowers, self-incompatibility, and a possible new type of mimicry involving males of Lepidoptera', submitted to the special issue of Plants. This is a very interesting paper that provides new information on the pollination and breeding system of Epidendrum densiflorum (Orchidaceae, Laeliinae). I believe that the authors' hypotheses and suggestions are reasonable and supported by the results of their observations.
I believe that this article should be published because it provides new and valuable information on the pollination biology of investigated orchid - E. densifolium. However, the paper needs minor corrections before publication.
(1) I ask the authors to consider changing the title, in the current version it is too long. (2) Please rewrite the Abstract, it should contain the most important results of the study, suggestions and hypotheses in my opinion should be given only in the Discussion section. (3) The Conclusion chapter should to synthesise all major points covered in your study and to tell the reader what they should take away from your work. You need to tell them what you found, why it’s valuable, how it can be applied, and what further research can be done. In this chapter, we do not provide hypotheses, but the most important research results. Also, this section needs to be rewritten.
I would like to congratulate the authors on the results obtained. Studying the biology of orchid pollination under natural conditions is very difficult, and observations of pollinators sometimes depend on luck. The video attached to the article confirms the conclusions of the work, the data presented does not raise any doubts. Excellent work, congratulations!
Author Response
Reviewer 2
I warmly acknowledge your suggestions/corrections that have certainly improved the manuscript. I also acknowledge your kind comments regarding the manuscript itself and the supplementary material (video). I have followed your suggestions as close as possible. Otherwise than indicated, I modified the text according to the suggestions/corrections you made. Modifications (including those suggested by another reviewer) are highlighted in yellow, to make the checking easier. Following your suggestions, the Title was shortened and the Hypothesis (originally in the Conclusion) was transferred to the Discussion (now lines 260-265). My only disagreement is the Abstract since I feel is important to briefly mention the Hypothesis there. We formulated that hypothesis as a result of our observations and I think it is important to explicitly enunciate the hypothesis we intend to test soon. On the other hand, I believe that keeping the hypothesis in the Abstract should render the text more interesting for potential readers. Thank you sincerely for your time and attention,
Prof. Rodrigo B. Singer (correspondent author)
